# Prevalence and Risk Factors for Post-Traumatic Stress in Hospitalized and Non-Hospitalized COVID-19 Patients

**DOI:** 10.3390/ijerph18042079

**Published:** 2021-02-20

**Authors:** Gunnar Einvik, Toril Dammen, Waleed Ghanima, Trond Heir, Knut Stavem

**Affiliations:** 1Pulmonary Department, Akershus University Hospital, 1478 Lørenskog, Norway; gunnar.einvik@medisin.uio.no; 2Institute of Clinical Medicine, University of Oslo, 0450 Oslo, Norway; waleed.ghanima@so-hf.no (W.G.); trond.heir@medisin.uio.no (T.H.); 3Department of Behavioral Medicine, Institute of Basic Medical Sciences, University of Oslo, 0317 Oslo, Norway; toril.dammen@medisin.uio.no; 4Haematology and Oncology, Østfold Hospital Trust Kalnes, 1714 Grålum, Norway; 5Norwegian Centre for Violence and Traumatic Stress Studies, 0484 Oslo, Norway; 6Health Services Research Unit, Akershus University Hospital, 1478 Lørenskog, Norway

**Keywords:** post-traumatic stress disorder, COVID-19, cohort study, outpatients, hospitalized, intensive care unit

## Abstract

This population-based study assessed the prevalence and determinants of symptom-defined post-traumatic stress disorder (PTSD) in a cohort of hospitalized and non-hospitalized patients about 1.5–6 months after their COVID-19 onset. The data were acquired from two mixed postal/web surveys in June–September 2020 from patients all aged ≥18 years with a positive polymerase chain reaction for severe acute respiratory syndrome Coronavirus-2 (SARS-CoV-2) until 1 June 2020, comprising both hospitalized and non-hospitalized subjects. The catchment areas of the two included hospitals covers about 17% of the population of Norway. In total, 211 hospitalized and 938 non-hospitalized subjects received invitation. The prevalence of symptom-defined PTSD was assessed using the PTSD checklist for DSM-5 (PCL-5). Determinants of symptom-defined PTSD and PTSD symptoms were analyzed using multivariable logistic and linear regression analysis. In total, 583 (51%) subjects responded at median 116 (range 41–200) days after COVID-19 onset. The prevalence of symptom-defined PTSD was 9.5% in hospitalized and 7.0% in non-hospitalized subjects (*p* = 0.80). Female sex, born outside of Norway, and dyspnea during COVID-19 were risk factors for persistent PTSD symptoms. In non-hospitalized subjects, previous depression and COVID-19 symptom load were also associated with persistent PTSD symptoms. In conclusion, COVID-19 symptom load, but not hospitalization, was associated with symptom-defined PTSD and PTSD symptom severity.

## 1. Introduction

The coronavirus disease 2019 (COVID-19) has to date affected more than 100 million people worldwide [1]. The proportion of cases in Norway having severe disease requiring hospitalization is about 5%, however, in the early phase of the pandemic, 13% of all cases were hospitalized [2]. Severe COVID-19 is characterized by dyspnea, chest pain, or confusion and 12% of hospitalized patients have critical illness requiring care in the intensive unit. In the general population, there is considerable concern about being infected [3], and social isolation and quarantine for several weeks or months may have important social and psychological impacts.

Post-traumatic stress disorder (PTSD) is a chronic and debilitating mental condition that may develop in response to catastrophic life events, including critical medical conditions. According to a meta-analysis, 17–44% of critical illness survivors report clinically important PTSD symptoms [4]. How COVID-19, with or without critical illness, impacts long-term PTSD is unknown, however, according to data from other medical conditions, the severity of the medical condition is likely to determine the future risk for PTSD. Studying symptoms of PTSD and their determinants after COVID-19 may therefore be important to inform about the prognosis of COVID-19, identify possible modifiable risk factors, and identify vulnerable patients to enable early intervention [5].

The few follow-up studies of survivors 1–3 months after COVID-19 hospitalizations have reported significant PTSD symptoms among 12–22%, as measured by the PTSD symptom checklist for DSM 5 (PCL-5) [6,7,8]. Non-hospitalized subjects constitute a much larger patient group than those hospitalized, and to the best of our knowledge, no previous study has assessed PTSD symptoms several months after COVID-19 in a consecutive sample of both hospitalized and non-hospitalized patients in a population-based study. Furthermore, studies on risk factors for the development of PTSD symptoms in patients after COVID-19 are scarce. It is, however, unknown whether the potential influence by COVID-19 on persistent PTSD symptoms is related to risk factors for PTSD in general (female sex, living alone, history of depression), COVID-19 symptom load, or the impact of hospitalization.

In this study, we aimed to determine if the prevalence of symptom-defined PTSD 1.5–6 months after confirmed COVID-19 was higher in hospitalized than non-hospitalized subjects. We also aimed to determine risk factors for persistent symptoms of PTSD in COVID-19 survivors.

## 2. Materials and Methods

### 2.1. Design and Population

In this study, we reported data from cross-sectional surveys of subjects in two parallel longitudinal cohort studies covering an identical geographical area of Norway representing 17% of the Norwegian population, conducted by the same research group.

#### 2.1.1. Survey of Hospitalized Subjects

The patient-reported outcomes and lung function after hospitalization for COVID-19 (PROLUN) study invited subjects 4–8 weeks after being discharged from a COVID-19-related hospital stay to participate in a follow-up study with electronic or paper questionnaires and a clinical visit about 3 months after discharge [9]. The present study includes only subjects discharged before 1 June 2020, and who resided in the catchment area of Akershus University Hospital and Østfold Hospital Trust Kalnes.

#### 2.1.2. Survey of Non-Hospitalized Subjects

The patient-reported outcomes and thromboembolic disease in COVID-19 (PROTROM) study recruited subjects residing in the catchment areas of Akershus University Hospital and Østfold Hospital Trust Kalnes that by 1 June 2020, had a positive polymerase chain reaction test for severe acute respiratory syndrome-Coronavirus-2 (SARS-CoV-2) from one of the two hospital laboratories or from the largest private laboratory in the region (Fürst laboratory). By the end of June 2020, 1–4 months after a positive test, 938 subjects were invited by mail for study participation. They could choose to respond to a paper or an online version of the questionnaire. After about 5 weeks, we sent a postal reminder to non-respondents. Details of the study design have previously been reported [10].

In total, 125/211 (59%) in PROLUN and 458/938 (49%) in PROTROM consented to participate, representing 51% of all eligible alive SARS-CoV-2 positive subjects >18 years in this geographical area and during this time period (Figure 1).

#### 2.1.3. Ethical Considerations

All participants signed a written or digital consent form prior to completing questionnaires. The Regional Committees for Medical and Health Research Ethics, Health Regon South East (PROLUN approval no.2020/125384, PROTROM approval no. 2020/149384) and the Data Protection Officer at Ahus approved both studies. PROLUN is registered in ClinicalTrials.gov (NCT04535154).

### 2.2. Data Collection and Assessments

#### 2.2.1. Assessment of PTSD

Symptoms of PTSD were assessed using the PCL-5 [11,12]. This questionnaire contains 20 items on an ordinal scale (0 to 4), which are aggregated into a total sum-score (range 0–80, with 80 denoting maximal symptom severity the past month). The PCL-5 also has an alternative scoring algorithm according to the DSM-5 criteria, requiring a score of ≥2 on 1 of 5 items in the Intrusion cluster, 1 of 2 in Avoidance, 2 of 7 in Cognition and mood, and 2 of 6 in Arousal and reactivity [13]. We used the DSM-5 scoring to define the caseness of PTSD based on self-reported symptoms. The Norwegian PCL-5 was developed through a translation-back-translation procedure and has been used in previous studies [14].

#### 2.2.2. Assessment of Comorbidity and COVID-19 Symptoms

In the survey of non-hospitalized subjects, comorbidity was self-reported using a checklist of 21 physician-diagnosed conditions, 18 of which constituted a self-report version of the Charlson comorbidity index [15] and additional items that we thought might be of interest in regard to the vulnerability for PTSD after COVID-19 including: history of depression, venous thromboembolism and more details on pulmonary comorbidity. In the hospitalized subjects, the Charlson comorbidity index [16] was collected by the review of electronic medical records (EMR) by trained physicians or nurses. Because of differences in data collection mode for comorbidities, we harmonized the comorbidities to represent the 16 most commonly occurring comorbidities/conditions. The number of these comorbidities was categorized as 0, 1, ≥2 comorbidities.

Non-hospitalized subjects retrospectively recorded the presence or absence of 23 symptoms related to COVID-19 [17]. In hospitalized subjects the presence or absence of COVID-19 symptoms were assessed by study physicians or nurses by reviewing the EMR related to the hospitalization. For those subjects the list of symptoms contained a few less symptoms than the study of non-hospitalized subjects, therefore the harmonization of all symptom variables between the two studies was not possible. From symptom assessments, we only included fever and dyspnea, because they were identically assessed and rated, as absent = 0 or present = 1, across the two studies.

### 2.3. Statistical Analyses

The characteristics of the participants are presented as the mean (SD), median (interquartile range or range) or number (%), as appropriate. We compared descriptive variables using the *t*-test for continuous variables, and chi-squared or Fisher’s exact test for categorical variables. Most missing values were for place of birth or marital status, which we considered difficult to impute. There were ≤2% missing values for other variables; therefore, we did not impute missing values.

The crude prevalence of PTSD according to DSM-5 was compared between groups using the chi-squared test. In addition, we present PTSD symptoms (0–80 range) as continuous scores. Determinants of symptom-defined PTSD were analyzed by multivariable logistic regression in the combined data set with common variables in the two samples. As we had 43 cases with symptom-defined PTSD, we estimated prior to analysis to include a maximum of five independent co-variables in the logistic regression models. In addition to hospitalized vs. non-hospitalized, we included the following independent variables in the models based on the literature and what we thought might be important: age (per decade), sex, living alone (married/cohabiting vs. single/divorced/widowed), born in Norway (yes vs. no) and time since COVID-19 onset (above vs. below the median of 116 days). All these variables were entered into the multivariable models without any statistical selection procedure.

Determinants of PTSD symptoms (continuous PCL-5 total scale) were analyzed using multiple linear regression analysis. Because the distribution of responses on this variable was highly skewed with many subjects having a score of 0, we attempted log and square-root transformations of the PCL-5 total scale. However, this did not improve the distribution of the residuals or the model fit. Therefore, we used multiple linear regression analysis of untransformed values, using bootstrapping with 10,000 iterations to estimate 95% confidence intervals. This analysis was conducted in the combined data set with common variables in the two samples (model 1); stratified for hospitalized (model 2) and non-hospitalized (model 3), in the non-hospitalized sample, with a richer set of variables (model 4).

In models 1, 2, and 3, we included age (per decade), sex, living alone (married/cohabiting vs. single/divorced/widowed), born in Norway (yes vs. no), time since COVID-19 onset (above vs. below the median of 116 days), education (three levels), number of comorbidities (0, 1, ≥2), and two symptoms (fever, dyspnea) in the models. Hospitalization vs. non-hospitalization was added only in model 1. Finally, for model 4 we replaced fever and dyspnea with the number of 23 self-reported acute COVID-19 symptoms categorized in tertiles; 0–5, 6–9, 10–23, and separately entered a symptom of confusion (as a possible marker of severe disease), as well as a history of depression (yes vs. no). There was no indication of collinearity, as assessed using Spearman correlations and variance inflation factor.

We used Stata (version 16.1, Stata Corporation, College Station, TX, USA) for all statistical analyses, using a significance level of *p* < 0.05 in two-sided tests.

## 3. Results

### 3.1. Study Population

The hospitalized subjects were older, comprised a larger proportion of men, had less education, and included a lower proportion of subjects born in Norway (Table 1). Moreover, hospitalized subjects had more asthma, hypertension, diabetes and several other comorbidities than non-hospitalized individuals, and they more often had dyspnea or fever during COVID-19.

### 3.2. Prevalence and Determinants of Symptom-Defined PTSD

PTSD, as indicated by the DSM-5 criteria, was present in 11 (9.5%) of hospitalized and 32 (7.0%) of non-hospitalized individuals (*p* = 0.80) (Table 2). Among women, 26/303 (8.6%) had PTSD compared with 17/268 (6.3%) among men (*p* = 0.31). The total PCL-5 score was higher in hospitalized than in non-hospitalized subjects, (mean 12.4 (SD 14.5) vs. 9.7 (11.3), *p* = 0.042). In multivariable logistic regression analysis, there was no association between being hospitalized and the presence of symptom-defined PTSD. Being born outside of Norway was associated with PTSD, but none of the other covariates were significantly associated with PTSD (Figure 2).

### 3.3. Determinants of the PTSD Symptom Scores

In the full sample data set, female sex, born outside Norway, and having dyspnea during COVID-19 were associated with higher PCL-5 total scores, while being hospitalized for COVID-19 was not (Table 3).

In stratified analyses, none of the independent variables was significantly associated with the PCL-5 total score in the hospitalized subjects. The pattern of associations with the PCL-5 total score was similar in the non-hospitalized sample as in the total sample with a few exceptions: living alone and having dyspnea during COVID-19 were associated with higher PTSD symptom scores (Table 3). In the analysis with the extended set of covariates among non-hospitalized subjects, male sex, being born in Norway, high symptom load during COVID-19 and confusions or reduced consciousness during acute COVID-19 were associated with higher PCL-5 total scores (Table 3). It is notable that time since COVID-19 onset was not associated with higher PCL-5 total scores in any of these four regression analyses.

## 4. Discussion

We investigated post-traumatic stress symptoms in a geographically defined population of hospitalized and non-hospitalized COVID-19-infected subjects on average 3 months after symptom onset. The prevalence of symptom-defined PTSD was 9.5% in hospitalized and 7.0% in non-hospitalized patients. Hospitalization was not associated with PTSD, while symptom load during COVID-19 added value to well known risk factors for determining the PTSD symptom score.

This is the first study that invited all positive SARS-CoV-2 subjects within a large geographical area during the first wave of the pandemic. Thus, our sample reflects a wide range of symptoms and trajectories of COVID-19 survivors.

Our prevalence rates are in the lower range of the 7–34%, which was reported in other cohorts of COVID-19 survivors following hospitalization at comparable time points [6,7,8,18,19,20,21], and much lower than the 96% reported during hospitalization in a study using the PCL-C, a civilian version of a PTSD questionnaire based on DSM-IV [22]. Two of the studies also comprised both hospitalized and non-hospitalized subjects [7,21]. Horn et al. assessed 180 subjects at about 7 weeks after COVID-19 onset also using the PCL-5 and reported comparable prevalence rates of PTSD as those in our study. De Lorenzo et al. assessed 185 subjects recruited in the emergency department and reported a prevalence of PTSD of 22% using the impact of event scale. One possible explanation for the varying prevalence is that studies use different definitions for caseness of PTSD, such as different cut-offs on a continuous symptom scale or DSM-5-based scores, as in the present study. Several other factors may also explain the large differences in prevalence estimates of reported PTSD between cohorts of COVID-19-infected individuals. These include the accessibility of adequate medical treatment in the healthcare system, people’s trust in health and welfare systems, and their access to factual or intimidating information. Compared with many other countries, Norway had sufficient capacity in a free healthcare system, low or no excess mortality from the pandemic, state financial support for the needy, and generally high confidence in the health and welfare schemes during the initial stage of the COVID-19 pandemic. Norwegians reported the highest perceived efficacy of governmental reactions to COVID-19 in a recent study in six countries [23]. Probably for the same reasons, Norway also had lower prevalence of pandemic-related stress reactions in the general population than many other countries [14].

In this study, the prevalence rates of PTSD, as well as the severity of PTSD symptoms after adjustment for comorbidity, did not differ between hospitalized and non-hospitalized patients, which deviates from previous studies reporting higher odds of PTSD in hospitalized than non-hospitalized patients [7,21]. On the one hand, hospitalization is likely to increase the perception of how critical the event is for people’s lives, which is usually associated with more post-traumatic stress [24,25]. The experience of how dangerous the infection can be may also lead to higher stress reactions [26,27]. On the other hand, hospitalization may also have been perceived as caring and safe, which may have reduced the stress reactions. Care that emphasizes information, calming, sense of safety, self-efficacy and hope may reduce the risk for developing PTSD [28]. During the first pandemic wave in Norway, the threshold for hospitalization might have been high, as home isolation was generally accepted by the patients, and the community health providers had daily telephone contact to monitor those in need for hospitalization. However, some may develop PTSD symptoms after a longer time since the event than the follow-up period of this study [29,30]. Therefore, prevalence rates and predictors of PTSD should be assessed after longer follow-up in future studies.

Regardless of hospitalization or not, COVID-19 symptom load was associated with higher levels of post-traumatic stress reactions. This makes sense since a higher symptom burden can result in a greater experience of threat. We find it likely that this is especially true for dyspnea which may be an anxiety- and fear-inducing symptom. However, we did not specifically assess threat perception in our study. In this study, female subjects, those born outside of Norway and those with a history of depression displayed the highest odds of having PTSD symptoms after 3 months. This supports previous findings of a higher risk of PTSD following COVID-19 among females and those with a history of psychiatric disease [7]. Citizens born outside Norway may have less access to information and less trust in the health care service or the authorities’ handling of the pandemic, and should be offered information and care that is adapted to their specific situation.

The prevalence of PTSD in the present study was at the level reported in populations after natural catastrophes and disasters, typically 5–10% [29] and in some general populations during the current pandemic of 3–8% [31,32,33]. However, the prevalence was smaller than that reported in a recent Norwegian study during the early stages of COVID-19 of 12.5% in men and 19.5% in women [14]. We believe this indicates that the generic fear of contracting COVID-19 and the social isolation and quarantine impacts PTSD symptoms as much as having the viral infection itself. Furthermore, some of the variation in prevalence rates could be attributed to the sampling method, where the studies with the highest prevalence recruited through public channels and social media, biased by persons with symptoms.

Our study has some limitations. The overall response rate in the non-hospitalized sample was about 50% and somewhat biased towards females and subjects >50 years of age. The response rate was low in three districts of Oslo with a high proportion of immigrants. This pattern of non-response is in line with our expectations and common in epidemiological surveys [34,35]. This may cause non-response bias of both the main variable and covariates, which is difficult to control for and may limit generalizability. It is possible that the elderly with much comorbidity and younger people with few problems, as well as many with limited Norwegian language skills, were less willing to respond, and this may overestimate the prevalence of PTSD. Furthermore, we did not include other relevant factors such as living with children, or death of a family member, that have been associated with PTSD risk in other COVID-19 studies [6], or general risk factors for PTSD such as details on previous psychiatric disease, although depression was included as a self-reported item in the non-hospitalized sample. Finally, COVID-19 symptoms were retrospectively reported and subject to recall bias, which may inflate actual symptoms.

In conclusion, persons infected during the first pandemic wave of COVID-19 in Norway had a lower prevalence of PTSD than those in previous reports. Being hospitalized for COVID-19 was not associated with higher prevalence of PTSD than was being infected with COVID-19 in general. Dyspnea and overall symptom load were associated with increased risk of persistent PTSD symptoms, in addition to being female, being born outside Norway and having a history of depression. These characteristics may serve to identify subjects that could benefit from follow-up with assessment for PTSD symptoms, regardless of having been hospitalized or not.

## Figures and Tables

**Figure 1 ijerph-18-02079-f001:**
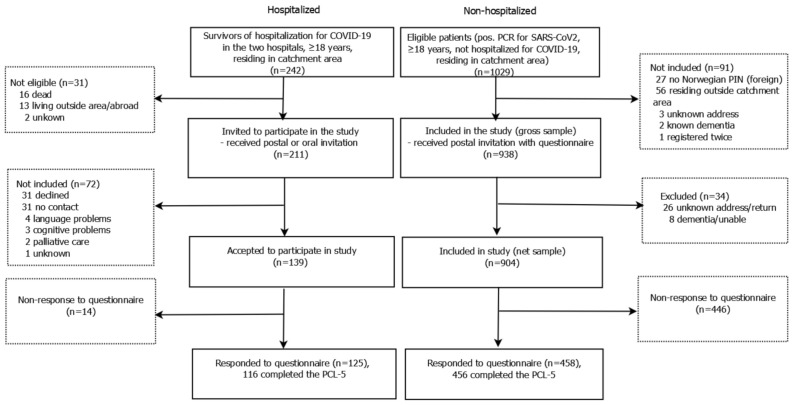
Flow chart of studies. PCL-5, post-traumatic stress disorder symptom checklist for DSM-5 (PCL-5); PCR, polymerase chain reaction; SARS-CoV-2, severe acute respiratory syndrome Coronavirus-2.

**Figure 2 ijerph-18-02079-f002:**
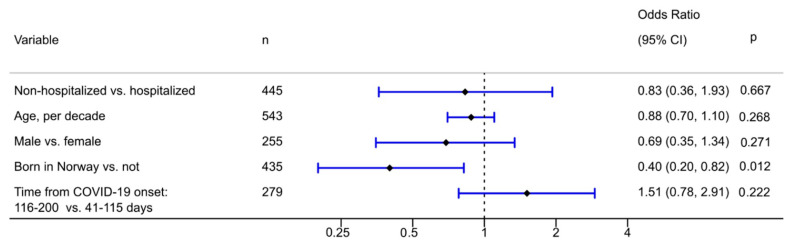
Odds ratios for the symptom-defined PTSD according to DSM-5 scoring with 95% confidence intervals and *p*-values, multivariable logistic regression analysis (*n* = 559).

**Table 1 ijerph-18-02079-t001:** Descriptive statistics for respondents in the two cohorts (*n* = 583), number (%) unless otherwise stated.

	Hospitalized (*n* = 125)		Non-hospitalized (*n* = 458)		*p*
	*n*		*n*		
Age, date of response, mean (SD)	125	57.7 (14.2)	458	49.6 (15.3)	<0.001
Sex, males	72	(58)	202	(44)	0.007
*Highest attained education (*n* = 123/458)*					<0.001
Primary school (<11 years)	33	(27)	41	(9)	
Secondary school (11–13 years)	43	(35)	174	(38)	
University (>13 years)	47	(38)	243	(53)	
*Marital status (*n* = 112/457)*					
Married/cohabiting	77	(69)	336	(74)	0.31
Born in Norway (*n* = 112/454)	70	(63)	382	(84)	<0.001
*Smoking status (*n* = 111/453)*					
Previous/current smoker	47	(42)	155	(34)	0.11
*Place of contraction (*n* = 125/456)*					0.001
Travel abroad	40	(32)	117	(26)	
In Norway, known contact	30	(24)	194	(43)	
In Norway, unknown contact	55	(44)	145	(32)	
*Time from COVID-19 onset, days, mean (SD)*	119	112 (30)	452	118 (27)	
*Time from COVID-19 onset, days (*n* = 119/452)*					0.006
41–115	72	(60)	209	(46)	
116–200	47	(40)	243	(54)	
*Comorbidity*					
Asthma	25	(20)	52	(11)	0.016
Chronic obstructive pulmonary disease	4	(3)	5	(1)	0.10
Other chronic lung		(0)	8	(2)	0.21
Lymphoma		(0)		(0)	-
Other cancer	4	(3)	6	(1)	0.23
Gastrointestinal	5	(4)	32	(7)	0.30
Heart problems	13	(10)	27	(6)	0.11
Hypertension	35	(28)	86	(19)	0.034
Circulatory	1	(1)	8	(2)	0.69
Chronic kidney disease	2	(2)	5	(1)	0.65
Liver disease	0	(0)	1	(0)	1
Neuromuscular	6	(5)	3	(1)	0.004
Stroke	4	(3)	8	(2)	0.30
Rheumatic	1	(1)	23	(5)	0.039
Venous thromboembolism	5	(4)	11	(2)	0.35
Diabetes	15	(12)	16	(3)	0.001
*Number of 16 comorbidities*					0.001
0	52	(42)	269	(59)	
1	42	(34)	124	(27)	
≥2	31	(25)	65	(14)	
Depression (*n* = 458) *		N/A	30	(7)	-
*Symptoms during acute COVID-19*					
Dyspnea (*n* = 125/449)	88	(70)	260	(58)	0.013
Confusion (*n* = 456) *		N/A	64	(14)	-
Fever (*n* = 125/453)	105	(84)	331	(73)	0.013

* Self-report, N/A or not comparable in hospitalized. SD, standard deviation.

**Table 2 ijerph-18-02079-t002:** Symptoms of post-traumatic stress (PCL-5) (*n* = 572).

	Hospitalized		Non-Hospitalized		*p*
	*n*		*n*		
PTSD (DSM-5 scoring), number (%)	116	11 (9.5)	455	32 (7.0)	0.80
PCL-5 total score (0–80 range), mean (SD)	116	12.4 (14.5)	456	9.7 (11.3)	0.042

PCL-5, PTSD symptom checklist for DSM-5; PTSD, post-traumatic stress disorder.

**Table 3 ijerph-18-02079-t003:** Determinants of symptoms of post-traumatic stress (PCL-5 total scores). Unstandardized beta coefficients with bootstrapped 95% confidence intervals and *p*-values, in stratified analyses. Multivariable linear regression.

	All		Hospitalized		Non-Hospitalized			
Model	(1)		(2)		(3)		(4)	
	Coef.	95% CI	Coef.	95% CI	Coef.	95% CI	Coef.	95% CI
*Cohort*								
Hospitalized	0							
Non-hospitalized	−1.05	[−4.38,2.28]						
*Age, per decade*	−0.31	[−0.98,0.36]	−1.87	[−4.14,0.40]	0.01	[−0.72,0.75]	0.01	[−0.70,0.73]
*Sex*								
Female	0		0		0		0	
Male	−3.01 **	[−5.01,−1.01]	−3.47	[−10.24,3.30]	−3.25 **	[−5.38,−1.13]	−2.69 **	[−4.67,−0.71]
*Born in Norway*								
No	0		0		0		0	
Yes	−6.82 ***	[−9.82,−3.82]	−5.64	[−12.97,1.68]	−7.17 ***	[−10.45,−3.89]	−5.14 **	[−8.59,−1.68]
*Highest attained education*								
Primary school (<11 years)	0		0		0		0	
Secondary school (11–13 years)	2.14	[−1.37,5.65]	2.57	[−4.90,10.03]	0.92	[−3.33,5.16]	1.69	[−1.97,5.35]
University (>13 years)	1.34	[−2.28,4.95]	5.75	[−2.41,13.91]	−0.58	[−4.74,3.58]	0.30	[−3.15,3.75]
*Marital status*								
Single/divorced/widowed	0		0		0		0	
Married/cohabiting	−1.27	[−3.40,0.85]	3.44	[−2.71,9.58]	−2.86 *	[−5.15,−0.57]	−1.78	[−4.03,0.47]
*No. of 16 comorbidities*								
0	0		0		0		0	
1	0.16	[−2.12,2.43]	1.17	[−6.54,8.89]	−0.10	[−2.41,2.20]	−0.99	[−3.05,1.07]
≥2	2.60	[−0.42,5.62]	3.76	[−3.60,11.12]	2.62	[−0.78,6.02]	1.87	[−1.66,5.41]
*Dyspnea during COVID-19*								
No	0		0		0			
Yes	3.68 ***	[1.80,5.57]	2.73	[−2.51,7.97]	3.66 ***	[1.58,5.73]		
*Fever during COVID-19*								
No	0		0		0			
Yes	1.80	[−0.26,3.86]	0.98	[−7.46,9.41]	1.70	[−0.39,3.78]		
*Time since COVID-19 onset, days*								
41–115	0		0		0		0	
116–200	0.63	[−1.40,2.66]	2.33	[−3.51,8.17]	0.59	[−1.49,2.68]	0.16	[−1.78,2.11]
*History of depression* ^1^								
No							0	
Yes							4.58 *	[0.19,8.97]
*No. of 23 symptoms during COVID-19* ^1^								
0–5							0	
6–9							0.59	[−1.74,2.93]
10–23							5.59 ***	[2.94,8.24]
*Confusion during COVID-19* ^1^								
No							0	
Yes							6.80 ***	[3.12,10.49]
*n*	531		98		433		444	
R-squared	0.11		0.13		0.13		0.24	

* *p* < 0.05, ** *p* < 0.01, *** *p* < 0.001. ^1^ only available in non-hospitalized. CI, confidence interval, PCL-5, PTSD symptom checklist for DSM-5.

## Data Availability

The data presented in this study are available on request from the corresponding author. The data are not publicly available due to local restrictions.

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
