# Peer review of "Prevalence and Risk Factors for Post-Traumatic Stress in Hospitalized and Non-Hospitalized COVID-19 Patients"

_ijerph, 2021, doi:10.3390/ijerph18042079_

Round 1

Reviewer 1 Report

This paper used a sample of COVID-19 patients and assessed the prevalence of PTSD and the risk factors of PTSD symptoms in this population through two cohort studies. It makes contributions to the understanding of the traumatic reactions in COVID-patients. However, the data analyses seem to be a bit simple, and I suggest some revisions so that the manuscript would be more informative. These are as follows.

Introduction

  1. The relevant findings on the risk factors for the development of PTSD symptoms in COVID-19 patients should be reviewed more thoroughly. Moreover, as the risk factors for PTSD, in general, have been explored in abundant literature, what makes special the examination of these factors in the background of COVID-19 should be stated elaborately to demonstrate the unique value of this study.

  1. Line 56, the time “after COVID-19” is somewhat ambiguous, and is needed to be described more accurately.

Methods and results

  1. Line 139, 140. The results of the two sets of multivariable logistic regression described here seem not to be shown in this paper.

  1. Why not choose fever and dyspnea, which were added to the multiple linear regression model for analyzing the determinants of PTSD symptoms, as an independent co-variable in the logistic regression model?

  1. As PTSD has four symptom clusters defined by DSM-5 which may have different manifestations in the pandemic, examining respectively the determinants of these clusters rather than the total score may be more informative.

Discussion

  1. Line 219, the description “covering 17% of the Norwegian population” is confusing, does it mean that the subjects account for 17% of the Norwegian?

  1. Line 232, the differences in the chosen cut-offs and definitions of PTSD should be specified.

  1. Line 272 to 281, it is not logical to place this paragraph here, as it mainly focused on the prevalence of PTSD which had been discussed above.

  1. The implications of the risk factors of PTSD symptoms should be discussed more in-depth.

Several minor errors appear.

e.g., Line 245, an extra space; Line 258, the typographical error “relayed”; Table 3, a typographical error “COVID-191” …… The authors need to check the paper thoroughly.

Author Response

Response to Reviewer #1:
This paper used a sample of COVID-19 patients and assessed the prevalence of PTSD

and the risk factors of PTSD symptoms in this population through two cohort studies. It

makes contributions to the understanding of the traumatic reactions in COVID-patients.

However, the data analyses seem to be a bit simple, and I suggest some revisions so

that the manuscript would be more informative. These are as follows.

  1. The relevant findings on the risk factors for the development of PTSD symptoms in

COVID-19 patients should be reviewed more thoroughly. Moreover, as the risk

factors for PTSD, in general, have been explored in abundant literature, what makes

special the examination of these factors in the background of COVID-19 should be

stated elaborately to demonstrate the unique value of this study.

Reply: Thank you for the opportunity to better describe the background for our study with regard to risk factors for PTSD after COVID-19. We have elaborated on why COVID-19 may be a risk factor for PTSD. However, data from other studies are relatively sparse regarding other risk factors in COVID-19 patients. All mentioned risk factors have been covered in the referenced articles.

  1. Line 56, the time “after COVID-19” is somewhat ambiguous, and is needed to be

described more accurately.

Reply; We have edited the sentence to “…1.5-6 months after confirmed COVID-19….” Line 59.

  1. Line 139, 140. The results of the two sets of multivariable logistic regression

described here seem not to be shown in this paper.

Reply: Thanks for pointing out this error. Indeed, for the outcome of PTSD symptom-based diagnosis, the multivariable analysis was performed in the complete sample, and this analysis is presented in Figure 2. We have edited the method section accordingly: “Determinants of symptom-defined PTSD (DSM-5 scoring) were analyzed by multivariable logistic regression in the combined data set with common variables in the two samples.” (Line 145-146)

  1. Why not choose fever and dyspnea, which were added to the multiple linear

regression model for analyzing the determinants of PTSD symptoms, as an

independent co-variable in the logistic regression model?

Reply: Again, thanks for the suggestion. We have handled the multivariable logistic and linear regression models differently according to the sample size. The number of independent variables that could be included in the regression model, in particular in the logistic regression model was limited. We followed a “rule-of-ten”, i.e. at least 10 cases per variable in logistic regression models. Although it would have been relevant and interesting to add symptoms to the model, we prioritized established risk factors for PTSD. Moreover, many of the symptoms were registered differently in the two studies and are not necessarily comparable. In hospitalized patients, symptoms were collected from electronic medical records, while in non-hospitalized patients, symptoms were self-reported in a questionnaire.

  1. As PTSD has four symptom clusters defined by DSM-5 which may have different

manifestations in the pandemic, examining respectively the determinants of these

clusters rather than the total score may be more informative.

Reply: Exploring determinants of each symptom cluster are interesting in the pandemic as well as in other crises, disasters or life events that cause post-traumatic stress. To investigate this would be to go even deeper into the field of trauma related reactions. The counter-perceptions are that the level of detail may become inappropriately high and that the totality of the understanding may become less clear. The usual approach in the trauma field is to study PTSD as one phenomenon consisting of three or four clusters. This applies to research on covid-19 as well as research on disasters or serious life events. Furthermore, for this research question the same limitations toward test abundance in a moderate-sized sample apply. We have thus chosen to stick to the same concept in this article, but are grateful for the idea for future research. For example, it may be interesting to investigate whether the pandemic produces a different composition of post-traumatic symptoms than other serious events.

  1. Line 219, the description “covering 17% of the Norwegian population” is confusing,

does it mean that the subjects account for 17% of the Norwegian?

Reply: We agree that this is not clear. 

We have revised the sentence in the revision: “In this study, we report data from cross-sectional surveys of subjects in two parallel longitudinal cohort studies covering an identical geographical area of Norway representing 17% of the Norwegian population, conducted by the same research group.” (line 65-66), and omitted this description in the discussion section: “This is the first study that invited all positive SARS-CoV-2 subjects within a large geographical area during the first wave of the pandemic.” (line 228)

  1. Line 232, the differences in the chosen cut-offs and definitions of PTSD should be

specified.

Reply: Thank you, we have edited this sentence to “One possible explanation for the varying prevalence is that studies use different definitions for caseness of PTSD, such as different cut-offs on a continuous symptom scale or DSM-5-based scores as in the present study.” (line 239-242).

  1. Line 272 to 281, it is not logical to place this paragraph here, as it mainly focused on

the prevalence of PTSD which had been discussed above.

Reply: We have changed some of the paragraphs in the discussion, but keep this paragraph here, as it discusses the external validation of the results (line 280).

  1. The implications of the risk factors of PTSD symptoms should be discussed more

in-depth.

Reply: Thank you for the important remark. We have tried to emphasize on this topic in the revised manuscript. rewritten the very last paragraph in the manuscript to meet this recommendation (line 277-279 + 303-304)

Several minor errors appear.

e.g., Line 245, an extra space; Line 258, the typographical error “relayed”; Table 3, a

typographical error “COVID-191” …… The authors need to check the paper thoroughly.

Reply: Thank you, we have checked the final revised version and hope to have excluded such errors. The COVID-191 is a formatting error, as it was COVID-191 (superscript 1, referring to a footnote)

Reviewer 2 Report

The manuscript is interesting and well-written. It focus on a particularly relevant topic, i.e., prevalence and predictors of PTSD in patients who were known to have had Sars-Cov-2. I have a few comments:

- Table 1: why is there a "?" for highest attained education?

  • In the discussion: you discuss the PTSD prevalence rates found in your study, which is in the lowest range of studies found in the literature.You argue that "Methodological differences may explain some of the differences between the studies, such as PTSD assessment methods, chosen cut-offs and definitions of PTSD, and the timing of assessments  after positive tests or hospital discharge.". Could you please discuss further these methodological differences and how they might have impacted the results?

Author Response

Response to Reviewer #2
The manuscript is interesting and well-written. It focus on a

particularly relevant topic, i.e., prevalence and predictors of PTSD in

patients who were known to have had Sars-Cov-2. I have a few

comments:

- Table 1: why is there a "?" for highest attained education?

Reply: We have removed this typographical error.

In the discussion: you discuss the PTSD prevalence rates found in

your study, which is in the lowest range of studies found in the

literature.You argue that "Methodological differences may explain

some of the differences between the studies, such as PTSD

assessment methods, chosen cut-offs and definitions of PTSD, and

the timing of assessments after positive tests or hospital

discharge.". Could you please discuss further these methodological

differences and how they might have impacted the results?

Reply: Thanks for this comment. Please see comment 7) to reviewer #1 which had the same question.

Reviewer 3 Report

The present manuscript entitled "Prevalence and risk factors for post-traumatic stress in 2 hospitalized and non-hospitalized COVID-19 patients" is an interesting article, whose objective is to determine if the prevalence of symptom-defined PTSD 1.5–6 months after COVID- 19 was higher in hospitalized than non-hospitalized subjects, and risk factors for persistent symptoms of PTSD in COVID-19 survivors.
Despite the topicality of the topic and the interest, it has certain limitations:

1. PTSD is mentioned in the abstract but the acronym is not defined. It would be convenient if it was described in the summary to be able to understand it when reading it.

2. The PCL scale is mentioned, but the acronyms are not defined.

3. In the Materials and Methods section, on line 61, the type of study carried out is not mentioned. The type of population is mentioned later, but it would be convenient to give more information in this section called "Design and population".

4. In relation to the ethical section, the approval by an ethics committee is mentioned, but the information given to patients, the information sheet and informed consent are not mentioned. How were you informed about the study? Were you mentioned everything related to the ethical section of your data and what it was going to be used for?

5. Both figures and tables lack a footnote indicating what the acronyms used mean.

6. Reference 1 is not well done. The same happens with the year in the others. It is necessary to review the norms of the magazine in relation to the appointments.

Author Response

Response to Reviewer #3
The present manuscript entitled "Prevalence and risk factors for post-traumatic stress in 2 hospitalized and non-hospitalized

COVID-19 patients" is an interesting article, whose objective is to determine if the prevalence of symptom-defined PTSD 1.5–6 months after COVID- 19 was higher in hospitalized than nonhospitalized subjects, and risk factors for persistent symptoms of

PTSD in COVID-19 survivors. Despite the topicality of the topic and the interest, it has certain limitations:

  1. PTSD is mentioned in the abstract but the acronym is not defined. It would be convenient if it was described in the summary to be able to understand it when reading it. 2. The PCL scale is mentioned, but the acronyms are not defined.

Reply: In the revised versions these acronyms are now defined in the text. (line 13, 16, 20)

  1. In the Materials and Methods section, on line 61, the type of study carried out is not mentioned. The type of population is mentioned later, but it would be convenient to give more

information in this section called "Design and population".

Reply: We have revised the first sentence explaining the design of the study: “In this study, we report data from cross-sectional surveys of subjects in two parallel longitudinal cohort studies covering an identical geographical area of Norway representing 17% of the Norwegian population” (line 64-66). The populations are further described in subsections 2.1.1 and 2.1.2, as we think it might be confusing for the readers using only one paragraph.

  1. In relation to the ethical section, the approval by an ethics committee is mentioned, but the information given to patients, the information sheet and informed consent are not mentioned. How were you informed about the study? Were you mentioned everything related to the ethical section of your data and what it was going to be used for?

Reply: Thank you for mentioning this topic. All participants were provided written information of the content and purpose of the study. This information letter had a signature page at the bottom, or a QR code to be scanned by a phone or tab to reach a secure digital form, hosted by the University of Oslo, to sign the consent. We have added a section 2.1.3 explaining ethical considerations; “All participants signed a written or digital consent form prior to completing questionnaires. (line 90-92)

  1. Both figures and tables lack a footnote indicating what the acronyms used mean.

Reply: We have added explanations of acronyms in all tables and figures.

  1. Reference 1 is not well done. The same happens with the year in the others. It is necessary to review the norms of the magazine in relation to the appointments.

Reply: The reference list has been updated according to the EndNote bibliography file.

Reviewer 4 Report

Specific comments:

32-33 where is the evidence that 10-30% have severe disease requiring hospitalisation? Most of the data indicates far fewer are hospitalised, and one feature of this pandemic is that the majority have few and non-severe symptoms. 

37-39 There is no real theoretical justification for expecting a link between Covid and PTSD. Covid is for most people not a 'critical illness'. The argument should perhaps differentiate between those hospitalised (and perhaps ventilated) and those not hospitalised.

55 Clarify that 'symptom-defined PTSD' is not an actual diagnosis of PTSD but is based on self-report.

84 'participated' - correct this 

119 Retrospective reporting of symptoms is likely to inflate actual symptoms experienced

188 No association between PTSD and hospitalisation, perhaps indicating that PTSD is not a result of Covid? Need to see the figures for the general population.

The overall pattern of results, with many statistical tests on a relatively small sample, may indicate that the findings have no real (eg clinical) significance other than statistical.

234-237 These are important points. It would have been helpful to have measured these variables.

258 'relayed' correct the word

Author Response

Response to reviewer #4

32-33 where is the evidence that 10-30% have severe disease requiring hospitalisation? Most of the data indicates far fewer are hospitalised, and one feature of this pandemic is that the majority have few and non-severe symptoms.

Reply: Thank you for this important comment. We agree that the total proportion of patients hospitalized is now lower than what stated in the manuscript. We think that this proportion was higher during the first pandemic wave, and in Norway the overall hospitalization rate was 13% between February and May 2020 (Nystad et al, Tidsskr. Nor Lægefor, 2020; 13). There was no limitation in hospital capacity. In some other countries, the capacity of hospital beds was probably a factor underestimating the proportion of patients with severe disease. However, as the test capacity has increased, the number of asymptomatic or mild disease has increased considerable. We thus agree to revise our statement, and refer to official Norwegian statistics which are complete: “The proportion of cases in Norway having severe disease requiring hospitalization is 5%, however, in the early phase of the pandemic, 13% of all cases were hospitalized.” (line 32-34)

37-39 There is no real theoretical justification for expecting a link between Covid and PTSD. Covid is for most people not a 'critical illness'. The argument should perhaps differentiate between those hospitalised (and perhaps ventilated) and those not hospitalised.

Reply: The first reports from China regarding PTSD post COVID-19 clearly state that the prevalence of PTSD may be high. Interestingly, Chinese studies found high prevalences of PTSD also in general population-based samples, not only hospitalized patients. We think that these observations form an empirical background for testing our study hypothesis. However, we agree that we have not detailed any psychological framework in which COVID-19 infection is a traumatic event in line with accidents, violence etc. We think that this is beyond the scope of a quantitative symptom query performed in our cohorts.

55 Clarify that 'symptom-defined PTSD' is not an actual diagnosis

of PTSD but is based on self-report.

Reply: We have clarified this in the revised version; We used the DSM-5 scoring to define caseness of PTSD based on self-reported symptoms (line 112-113).

84 'participated' - correct this

Reply: Corrected.

119 Retrospective reporting of symptoms is likely to inflate actual

symptoms experienced

Reply: We have added this sentence to the limitation-section: COVID-19 symptoms were retrospectively reported, which may inflate actual symptoms. (line 303-304)

188 No association between PTSD and hospitalisation, perhaps indicating that PTSD is not a result of Covid? Need to see the figures for the general population. The overall pattern of results, with many statistical tests on a relatively small sample, may indicate that the findings have no real (eg clinical) significance other than statistical.

Reply: Thank you for this comment. In our discussion we comment on that our reported prevalence is lower than several other studies reporting from the first period of the pandemic, and we state that risk factors present before COVID-19 are as well important risk factors for post COVID-19 PTSD symptoms. However, we also think that the association between COVID-19 symptoms and PTSD symptoms may be clinically relevant. COVID-19 symptoms will thus be an indicator of the intensity of COVID-19, and constitute a more sensitive marker among population-based samples in which the majority of participants have not been admitted to hospital. We think that e.g. one-year assessment in a large cohort of patients will be relevant to discuss this association further.

234-237 These are important points. It would have been helpful to have measured these variables.

Reply: We agree, but these variables were not quantifiable in our study.

258 'relayed' correct the word

Reply: Corrected

Reviewer 5 Report

The manuscript topic is actual and the paper has merit. It could be attractive, adequate and interesting for the journal readers. However there are some point that authors should address in order to have a final more complete paper. Epidemiological data are important, and it could be fundamental for the management such a situation in other area of the world related to the pandemic.

More data should be added in order to compare previous study and recent data as follows. A deep revision of the literature should be performed.  

Introduction section is poorly written. A paragraph about recent paper about COVID telemedicine and surface persistence of the virus may be added the beginning of the paper in order to have a more complete details about the SARS COV2. 
Something about Virus characteristic on surface and also about current trends and devices used for prevention should be added.
Fiorillo, L.; et al.. COVID-19 Surface Persistence: A Recent Data Summary and Its Importance for Medical and Dental Settings. Int. J. Environ. Res. Public Health 2020, 17, 3132
Cavallo, L.;  et al. . 3D Printing beyond Dentistry during COVID 19 Epidemic: A Technical Note for Producing Connectors to Breathing Devices. Prosthesis 2020, 2, 46-52.

Therefore some suggestions also about telemedicine and COVID and about the last published data should be added.
Cervino et al Covid 19 pandemic and telephone triage before attending medical office:problem or opportunity? Medicina may 2020; 56(5) :250
Cervino, G.; et al. . SARS-CoV-2 Persistence: Data Summary up to Q2 2020. Data 2020, 5, 81

Author Response

Response to reviewer #5

The manuscript topic is actual and the paper has merit. It could be attractive, adequate and interesting for the journal readers. However there are some point that authors should address in order to have a final more complete paper. Epidemiological data are important, and it could be fundamental for the management such a situation in other area of the world related to the pandemic.

More data should be added in order to compare previous study and recent data as follows. A deep revision of the literature should be performed.  

Introduction section is poorly written. A paragraph about recent paper about COVID telemedicine and surface persistence of the virus may be added the beginning of the paper in order to have a more complete details about the SARS COV2. 
Something about Virus characteristic on surface and also about current trends and devices used for prevention should be added.
Fiorillo, L.; et al.. COVID-19 Surface Persistence: A Recent Data Summary and Its Importance for Medical and Dental Settings. Int. J. Environ. Res. Public Health 2020, 17, 3132
Cavallo, L.;  et al. . 3D Printing beyond Dentistry during COVID 19 Epidemic: A Technical Note for Producing Connectors to Breathing Devices. Prosthesis 2020, 2, 46-52.
Therefore some suggestions also about telemedicine and COVID and about the last published data should be added.
Cervino et al Covid 19 pandemic and telephone triage before attending medical office:problem or opportunity? Medicina may 2020; 56(5) :250
Cervino, G.; et al. . SARS-CoV-2 Persistence: Data Summary up to Q2 2020. Data 2020, 5, 81

Reply: Thank you for the suggestions for discussion of our study in relation to other studies. We have read with interest the named articles regarding viral characteristics on surfaces, and for telemedicine strategies in relation to medical consultations. Both are highly relevant for medical care per se, and telemedicine may even be one strategy to keep patients calm during COVID-19. However, we are not fully convinced that these articles are directly relevant for the discussion of the two main aims of this study which are prevalence of PTSD symptoms and risk factors for such.

Round 2

Reviewer 1 Report

Thank you to the authors for their responses to my feedback on the revised version of the paper. I am satisfied with the current revised manuscript. However, the authors still need to check the paper thoroughly to eliminate minor errors, such as Line 66 where ":" should be a ".".

Author Response

Reviewer #1

Thank you to the authors for their responses to my feedback on the revised version of the paper. I am satisfied with the current revised manuscript. However, the authors still need to check the paper thoroughly to eliminate minor errors, such as Line 66 where ":" should be a ".".

Author response: Thank you for your willingness to review and improve our paper. We have performed additional spell-check and grammar-check in this revision throughout the paper.

Reviewer 4 Report

This manuscript is much improved, though I still have a number of issues that need to be resolved, mainly relatively minor, though perhaps the conclusion should be clearer that there is no apparent link between Covid and PTSD (which is a good thing).

l42 - what past data?Presumably not relating to Covid?

l54-57 or there is no relationship between Covid and PTSD

l86-88 Limited response rate. We do not know the characteristics of those who did not respond. If there is a link between Covid and PTSD then we do not know whether those with PTSD are more or less likely to respond

l197-198 Higher levels of PTSD in those born outside Norway - are they refugees with PTSD from their experiences?

l208-210 These findings perhaps suggest the PTSD levels found were not related to Covid, which illustrates the need for population PTSD figures

l277-279 See earlier point re there may be PTSD in these groups for other reasons, eg being a refugee. It is also likely that being in an alien culture (different language, different food) has an impact.

Overall it appears that reported PTSD levels are not higher in Covid patients than the population. This is an important finding as it shows Covid is not related to PTSD (draw a positive finding from a negative finding!). One problem with this is that now Covid has been around for about a year there may be a build up of anxieties leading to mental health issues that wasn't present in June, so the picture might be different now.

Author Response

Reviewer #4

This manuscript is much improved, though I still have a number of issues that need to be resolved, mainly relatively minor, though perhaps the conclusion should be clearer that there is no apparent link between Covid and PTSD (which is a good thing).

Author response: Thank you for your positive response. We have responded to each comment below, including revision of the conclusion:

In conclusion, persons infected in the first pandemic wave of COVID-19 in Norway had a lower prevalence of PTSD than those in previous reports. (lines 304-305)

l42 - what past data?Presumably not relating to Covid?

Author response: Yes, your presumption was correct. We have revised the paragraph:

How COVID-19, with or without critical illness, impacts long-term PTSD is unknown, yet according to data from other medical conditions, the severity of the medical condition is likely to determine the future risk for PTSD (lines 43-45)

l54-57 or there is no relationship between Covid and PTSD

Author response: We acknowledge this possibility; however, we think that the current statement is not tendentious toward an association.

l86-88 Limited response rate. We do not know the characteristics of those who did not respond. If there is a link between Covid and PTSD then we do not know whether those with PTSD are more or less likely to respond

Author response: We have revised somewhat in the section on limitations in the discussion to mention this topic:

The overall response rate in the non-hospitalized sample was about 50% and somewhat biased towards females and subjects >50 years of age. The response rate was low in three districts of Oslo with a high proportion of immigrants. This pattern of non-response is in line with our expectations and common in epidemiological surveys. This may cause non-response bias of both the main variable and covariates, which is difficult to control for and may limit generalizability. (line 290-295)

l197-198 Higher levels of PTSD in those born outside Norway - are they refugees with PTSD from their experiences?

Author response: The PCL-5 questionnaire is specific for one particular stressor; in this case the introductory text in PCL-5 instructed the participants to consider their symptoms in relation to their COVID-19 disease. We agree that the stress level prior to COVID-19 would have been higher if PTSD was already present. We do not whether some of the immigrants had a background as refugees or previous PTSD. The lack of objective information of pre-COVID-19 psychiatric disease is already mentioned in the limitation.

l208-210 These findings perhaps suggest the PTSD levels found were not related to Covid, which illustrates the need for population PTSD figures

Author response: Thank you for this important comment, we agree that this is a possible interpretation of the finding. Indeed, another Norwegian study (Bonsaksen et al) found a higher prevalence of PTSD in 2020 than in 2018. However, the latter study was not generalizable to a general population in line with the current study, and a head-to-head comparison of data is not applicable.

l277-279 See earlier point re there may be PTSD in these groups for other reasons, eg being a refugee. It is also likely that being in an alien culture (different language, different food) has an impact.

Author response: Thanks again, in line with the comment above pre-COVID-19 mental health may be one explanatory factor. From our clinical practice we know that some of the families with origin outside Norway had many family members that were affected by COVID-19 in. Worrying for sick family members, in addition to own disease, may be a factor. However, we have no systematic information on this in the data set, and therefore do not have a feasible variable for further analysis.

Overall it appears that reported PTSD levels are not higher in Covid patients than the population. This is an important finding as it shows Covid is not related to PTSD (draw a positive finding from a negative finding!). One problem with this is that now Covid has been around for about a year there may be a build up of anxieties leading to mental health issues that wasn't present in June, so the picture might be different now.

Author response: Please see our response to comments above concerning these topics.

Reviewer 5 Report

Authors didi not address the reviewer's notes and requests. Introduction  section is weak and poorly written. Not significant changes have been made comparing the first round of revision.

Author Response

Reviewer #5

Authors didi not address the reviewer's notes and requests. Introduction  section is weak and poorly written. Not significant changes have been made comparing the first round of revision.

Author response: Our response in the last revision contained our arguments for why the reviewer’s requests were not met. By our best intention we cannot see that the comment above contain any new information concerning the scientific rationale of adding the requested references. Thus, we have not performed additional editing.